# Metabolic Remodeling and Implicated Calcium and Signal Transduction Pathways in the Pathogenesis of Heart Failure

**DOI:** 10.3390/ijms221910579

**Published:** 2021-09-30

**Authors:** Antoine H. Chaanine

**Affiliations:** 1Department of Medicine, Heart and Vascular Institute, Tulane University, New Orleans, LA 70112, USA; achaanine@tulane.edu; Tel.: +1-(504)-988-1612; 2Department of Physiology, Tulane University, New Orleans, LA 70112, USA

**Keywords:** heart failure, mitochondria, metabolic remodeling, calcium, signaling

## Abstract

The heart is an organ with high-energy demands in which the mitochondria are most abundant. They are considered the powerhouse of the cell and occupy a central role in cellular metabolism. The intermyofibrillar mitochondria constitute the majority of the three-mitochondrial subpopulations in the heart. They are also considered to be the most important in terms of their ability to participate in calcium and cellular signaling, which are critical for the regulation of mitochondrial function and adenosine triphosphate (ATP) production. This is because they are located in very close proximity with the endoplasmic reticulum (ER), and for the presence of tethering complexes enabling interorganelle crosstalk via calcium signaling. Calcium is an important second messenger that regulates mitochondrial function. It promotes ATP production and cellular survival under physiological changes in cardiac energetic demand. This is accomplished in concert with signaling pathways that regulate both calcium cycling and mitochondrial function. Perturbations in mitochondrial homeostasis and metabolic remodeling occupy a central role in the pathogenesis of heart failure. In this review we will discuss perturbations in ER-mitochondrial crosstalk and touch on important signaling pathways and molecular mechanisms involved in the dysregulation of calcium homeostasis and mitochondrial function in heart failure.

## 1. Introduction

The heart is an organ with high-energy demands necessary to maintain its continuous mechanical work [1]. The mitochondria, which are the powerhouse of the cell, are most abundant in the heart [2]. Their role extends beyond adenosine triphosphate (ATP) production to include ion homeostasis, participation in calcium signaling and redox balance, and are considered the hub of cell metabolism [3]. This multiplicity of mitochondrial (mt)-function is tightly coordinated to maintain cell survival, otherwise cell death ensues [4]. Perturbations in mt-homeostasis and metabolic remodeling constitute important pathophysiological processes in the development of heart failure (HF). To date, it is well known that fatty acid (FA) metabolism and mt-FA-β-oxidation are attenuated in compensated hypertrophy and progressively worsen upon progression to overt systolic heart failure (SHF) [1,5]. Glucose metabolism is initially up-regulated in compensated hypertrophy then is attenuated in progression to SHF [1,6]. Oxidative phosphorylation (OXPHOS) is attenuated at the onset of left ventricular systolic dysfunction and drop in ejection fraction, and is impaired further upon progression to overt SHF [7]. Molecular mechanisms and signal transduction pathways regulating mt-function and cardiac metabolism are partially known to date. The peroxisome proliferator-activated receptor gamma coactivator 1-alpha (PGC-1α) and the peroxisome proliferator-activated receptor alpha (PPARα) signaling, which gradually decline in progression to overt SHF [8,9], are important players in the regulation of metabolism, primarily FA metabolism [10]. Protein kinase A (PKA) and its opponents protein phosphatase 1 (PP1) and calcineurin (PP2A) as well as the adenosine-monophosphate-activated protein kinase (AMPK) signaling, their interplay, play an important role in the regulation of cardiac metabolism and mt-function. Other kinases, such as protein kinase C epsilon (PKCε), calcium/calmodulin kinase II isoform delta (CaMKIIδ) and the Jun-N-terminal kinase (JNK), also modulate mt-function and dynamics, as discussed later in Section 5. It is not known which of these pathways predominate and what is the sequence of derangements in signal transduction pathways that lead to the initiation and progression of HF. This derangement in signaling is also paralleled by derangements in calcium cycling [11] and calcium-mediated endoplasmic reticulum (ER)–mt crosstalk; thereby, leading to metabolic reprogramming and mt-dysfunction in a milieu of heightened oxidative stress that constitutes the hallmark of the syndrome of HF. The topology of the intermyofibrillar mitochondria in the heart places them at the hub of calcium signaling as they are neighbored, at a very close proximity, by the T-tubules, the longitudinal sarcoplasmic reticulum (SR) and the sarcomeres as we discuss below. In this review, we will focus on calcium signaling in the regulation of mt-function and metabolism, and on signal transduction pathways governing metabolic remodeling and calcium cycling and homeostasis dysregulation in HF, which in our opinion, are two intertwined processes in the heart.

## 2. Mitochondrial Morphology, Topology and Function

Mitochondria are membrane bound organelles that are thought to have originated from symbiotic ancestors [12] and are found in almost all eukaryotic cells. They are comprised of an outer (OMM) and an inner mt-membrane (IMM). Between the OMM and IMM is the inter-membrane space (IMS) where the proton pump gradient forms across the IMM and is the sole determinant of mt-capacity for ATP production. The mt-matrix is surrounded by the IMM and is the site where most reactions related to metabolism and OXPHOS takes place. The mt-matrix also contains mt-DNA and the necessary mt-translation/elongation machinery that encodes 13 proteins that are essential for respiratory chain function in human [13,14]. However, the majority of mt-genes are encoded in the nucleus and are then imported into the mitochondria through a specialized mt-import and assembly systems [15].

There are three mt-subpopulations in the cardiac myocyte: the perinuclear, the intermyofibrillar and the subsarcolemmal mitochondria [16], Figure 1A. Of these, the intermyofibrillar mitochondria are most abundant and are most important due to their participation in calcium signaling and ATP production [17]. Each intermyofibrillar mitochondrion extends from one Z band to another and is surrounded by sarcomeres at each side and T-tubules at each end, Figure 1B. The longitudinal SR forms a network around the intermyofibrillar mitochondria and is tethered to them via tethering complexes [18], known as the ER-mitochondria encounter structure (ERMES) [15]. Thus, the OMM acts as a nodal membrane that connects the mitochondria with their surrounding and is the site where the majority of kinases and phosphatases are anchored to modulate mt-morphology and function and metabolism. Figure 2 is a schematic drawing summarizing the metabolic pathways, calcium cycling and implicated signaling regulating metabolism, calcium cycling and mt-function in the heart under normal physiological conditions.

Under normal conditions, mitochondrial ATP is generated primarily through the oxidation of fatty acid (60–90%) and glucose (10–40%). These substrates are processed in the cytoplasm and then are transported into the mitochondria as metabolic intermediates where they enter the tricarboxylic acid (TCA) cycle to produce reduced nicotinamide adenine dinucleotide (NADH), and flavin adenine dinucleotide (FADH2). NADH and FADH2 are then oxidized to NAD and FAD by the electron transport chain (ETC) system to generate a proton gradient across the IMM. The proton gradient is the driving force for ATP synthesis that takes place when protons flow via the ATP synthase (ETC complex V) into the mt-matrix. In the process, mt-reactive oxygen species (ROS) are physiologically generated, mainly from ETC complexes I and III in the form of oxygen radicals, which are reduced to hydrogen peroxide by the mitochondrial manganese superoxide dismutase. Hydrogen peroxide is then reduced to water by the mitochondrial or cytosolic antioxidant systems such as catalase, thioredoxin-peroxidase and glutathione-peroxidase. Calcium is a key second messenger that orchestrates the interplay of mt-redox and energetics with the excitation contraction coupling [19]. Increases in cytosolic and mt-calcium transients have been observed upon acute stimulation of the cardiomyocyte with isoproterenol and increases in energetic demand. This serves as a stimulus to activate enzymes in the TCA cycle and ETC complexes, such as pyruvate dehydrogenase, NAD^+^-dependent isocitrate dehydrogenase, α-ketoglutarate dehydrogenase and ETC complexes I and III [20], respectively. Therefore, the same signal that stimulates muscle contraction also stimulates ATP production to meet the higher energetic needs/demands, but at the expense of increased ROS generation. Crosstalk exists between calcium and ROS signaling systems. Together, they can regulate and fine tune cellular signaling networks [21]. Derangements in either of the aforementioned signaling systems would adversely affect the other and will have detrimental consequences on cellular fate and survival [21].

## 3. Calcium Signaling and Mitochondrial Oxidative Capacity

Calcium is an important second messenger involved in multiple cellular processes. It is crucial for the excitation-contraction coupling due to the calcium induced calcium release (CICR) phenomenon from the SR via the ryanodine receptors isoform 2 (RYR2) [22]. This phenomenon happens upon stimulation of the cardiomyocyte and opening of the L-type calcium channels (LTCC) leading to entry of small amount of calcium that triggers a large amount of calcium to be released from the junctional SR via RYR2 [23,24]. Then, calcium is pumped back into the SR via the enzyme sarco/endoplasmic reticulum ATPase (SERCA2a), promoting the initiation of the relaxation phase of the cardiac cycle, Figure 2. SERCA2a activity is regulated by phospholamban (PLN), which inhibits its activity [25,26]. The SR/ER and the mitochondria are important component of cardiac myocyte calcium cycling. They crosstalk with each other via calcium signaling through structural and functional interorganellar coupling [27]. The role of calcium in the ER is to control the calreticulin/calnexin cycle, which is required for proper protein folding, and cellular growth and development [28]. In mitochondria, calcium acts as a second messenger required to regulate enzymes of the TCA cycle and ETC complexes I and III. Calcium uptake into the mitochondria is orchestrated via the mostly abundant voltage dependent anion channel isoform 1 (VDAC1), compared to isoforms 2 and 3, at the OMM and is tightly regulated by the mitochondrial calcium uniporter (MCU) at the IMM [29,30], Figure 2. The MCU is a nodal regulator of calcium uptake into the mt-matrix and thus regulates mt-energetics and OXPHOS under cardiac stress conditions of high energetic demands [31]. Under basal conditions it seems that calcium can still enter the mt-matrix via MCU independent pathways to maintain cellular energetics, such as the ryanodine receptor isoform 1 (RYR1), however, this has not been well studied, Figure 2. VDAC1 is also an important player in mt-energetics/metabolism via its regulation of metabolites exchange across the mt-membranes and the ETC [32,33]. mt-calcium efflux is mediated via the sodium/calcium/lithium exchanger (NCLX) at the IMM into the IMS and then through VDAC1 into the cytoplasm [34]. Moreover, recent investigation has eluded to the importance of the hydrogen/calcium exchanger at the IMM, also known as the leucine zipper EF-hand containing transmembrane protein 1 (LETM1), to play an important role in regulating mt-calcium flux and energetics independent of MCU and NCLX [35]. Thus, the ER-mt crosstalk constitutes the functional unit that links calcium cycling to cardiac energetics. This ER-mt crosstalk relationship is tightly coordinated to maintain proper cellular function and is regulated by signal transduction pathways and B-cell lymphoma-2 (Bcl-2) family proteins at the ER-mt interface, Figure 2.

The ER-mt crosstalk relationship is perturbed in HF due to an alteration in cardiomyocyte ultrastructure and an imbalance in signaling and Bcl-2 family proteins regulation at the ER-mt interface. These changes take place at different stages of HF. Abnormal clustering of fragmented mitochondria is seen in pathological hypertrophy [36]. Upon transitioning to a decompensated state and SHF development, clustered fragmented mitochondria undergoing different stages of vacuolar degeneration are often seen and constitute the ultrastructural signature of HF [37,38], Figure 3. T-tubule dilatation and abnormal T-tubule-mt relationship [38] may alter calcium release from the dyadic cleft and the ER-mt crosstalk [39], Figure 3. The MAPK, JNK, which is activated in early pathological hypertrophy [36], plays detrimental role by promoting ER-mt calcium homeostasis dysregulation and apoptosis through the phosphorylation of Bcl-2 family proteins [36] and mitofusin 2(MFN2) [40]. Phosphorylation of PLN at Ser16 and dynamin related protein 1 (DRP1) at Ser637 were attenuated in a rat model of pressure overload induced moderate remodeling and early systolic dysfunction [9,41]. This leads to decreased calcium cycling and enhanced mt-fission and fragmentation, respectively, at the initial stages of cardiac remodeling and systolic dysfunction. It is not yet clear to whether these changes in PLN and DRP1 phosphorylation are related solely to an enhanced calcineurin activity [42] or also to an early decrease in PKA activity at the ER-mt interface subcellular compartment [43]. Eventually, the down-regulation of SERCA2a constitutes the signature for the progression to SHF and impaired calcium cycling [44,45] leading to depletion in ER calcium content and increase in cytosolic calcium, in addition to ER calcium leak via the RYR2 [46], Figure 4. Mitochondrial-matrix calcium overload and increased mt-ROS production are now being realized as important pathophysiological processes of mt-dysfunction and decreased oxidative capacity in HF [34,47,48]. It has been shown that MCU expression level and mt-calcium uptake were attenuated in diabetic cardiomyopathy [49]. This paradox in decreased MCU expression level and mt-matrix calcium overload could be explained by activation of signaling pathways that can modulate mt-calcium uptake and/or efflux proteins at the OMM and IMM via a post-translational modification (PTM) mechanism, Figure 4. This area may need to be elucidated in future studies. Moreover, mt-calcium efflux kinetics is a much slower process than mt-calcium uptake [50,51], owing to the possible explanation of increases in mt-matrix calcium during acute, such as isoproterenol stimulation, and chronic cardiac stress conditions, as is in HF. In the former, increases in mt-matrix calcium acts as a stimulant for ATP production; in the latter, mt-matrix calcium overload is detrimental leading to mt-vacuolar degeneration and increased ROS and apoptosis. These differences between acute vs. chronic cardiac stress are likely related to distinct signaling pathways and molecular mechanisms modulating mt-calcium homeostasis and mt-function that are not entirely known. Future work is needed in this area to elucidate molecular mechanisms regulating mt-calcium and metabolism in acute versus chronic cardiac stress.

Previous work showed that recombinant expression of VDAC1 enhanced ER-mt contact sites and calcium transfer into the mitochondria and apoptosis [52,53]. In HF, VDAC1 expression does not change compared with normal heart. However, VDAC1 oligomerization or modulation by the Bcl-2 family proteins [48], as discussed below, or in response to apoptotic stimuli [54], may alter calcium entry into the IMS and subsequently into the mt-matrix leading to mt-matrix calcium overload, mt-dysfunction and apoptosis, Figure 4. The exact molecular mechanism that promotes VDAC1 oligomerization in HF is not known. Moreover, previous work has shown that calcium uptake via MCU is enhanced by its PTM by the mt-CaMKII [55,56], Figure 4. Conditional NCLX knockout in mouse heart has been shown to be lethal, with less than 13% of affected animals surviving beyond day 14 [57]. These findings point to the importance of the NCLX in mt-calcium efflux. Recently, it has been shown that MCU over-expression rescues inotropy and reverses HF by reducing SR calcium leak [58]. Overall, these findings hint to the complexity of ER-mt calcium homeostasis regulation and function. Whether other modes of mt-calcium entry or other mechanisms regulating mt-calcium uptake and efflux at the IMM exist, need to be elucidated in future studies.

Previous work has shown that mt-biogenesis and oxidative capacity were preserved in patients with HF and preserved ejection fraction (HFpEF) [38] or enhanced in pathological compensated hypertrophy [9,59]. A decline in mt-content and oxidative capacity has been observed in rodent models upon transitioning from a compensated state to moderate remodeling and early systolic dysfunction [9]. At the molecular level, these findings are mirrored by a decrease in PGC-1α expression upon transitioning from compensated hypertrophy to early systolic dysfunction [9]. Cytochrome c oxidase activity is attenuated in early systolic dysfunction along with a decrease in expression of the ETC complexes I and IV [41]. Transitioning to overt SHF and HFrEF is marked by a further decrease in mt-biogenesis and PGC-1α expression and signaling, along with a decrease in expression of the ETC complexes, Figure 4. These findings were observed in an animal model of SHF and in human HFrEF [38,59]. Another mechanism of decreased oxidative capacity in HF is phosphorylation (discussed below) and hyperacetylation of mt-proteins. Increased acetylation of mt-proteins involved in mt-FA-β-oxidation, TCA cycle and ETC complexes has been observed in the early stages of HF in a pressure overload mouse model of transverse aortic constriction (TAC) and in end-stage human HFrEF [60,61]. Hyperacetylation of mt-proteins is speculated to be related to the presence of excessive acyl-CoAs and reduced protein deacetylation by the sirtuin family of NAD^+^-dependent deacetylases, mainly SIRT3 [62], and SIRT5. SIRT3 is the predominant mt-deacetylase isoform in the heart. SIRT3^-/-^ mice develop contractile dysfunction at 24 weeks of age [63]. Myocardial SIRT3 expression was increased in cardiac hypertrophy following TAC in mice [64] and then was downregulated at HF development [65,66]. Similarly, SIRT3 expression was decreased in the human failing heart [67]. PGC-1α was found to regulate SIRT3 expression through the AMPK-PGC-1α-SIRT3 axis in skeletal muscle [68]. Moreover, SIRT3 downregulation seems to coincide with the decrease in PGC-1α expression in rodent and human HF [69]. Thus, a decline in mt-oxidative capacity in early systolic dysfunction is mainly related to an increase in mt-ROS and a decrease in activity of the OXPHOS machinery. Transitioning to overt SHF is accompanied by decreases in mt-content and mt-biogenesis, including expression of the ETC complexes, as well as by an increase in mt-ROS and PTM, therefore, phosphorylation and hyperacetylation, of the OXPHOS machinery.

## 4. Bcl-2 Family of Proteins and Their Implication in Calcium Homeostasis and Metabolism

The Bcl-2 family of proteins are well known to control ER-mt calcium homeostasis, mt-outer membrane permeabilization (MOMP) and apoptosis [70,71]. Many Bcl-2 family proteins possess a C-terminal hydrophobic trans-membrane domain enabling them to be anchored to ER and mitochondria [72]. They are globular proteins containing α-helixes and are characterized by the presence of 1–4 Bcl-2 homology (BH) domains. They are classified as anti-apoptotic and pro-apoptotic Bcl-2 proteins based on the number of BH domains that they have. Anti-apoptotic Bcl-2 proteins (Bcl-2, Bcl-xL and Mcl-1) contain four BH domains, while the pro-apoptotic Bcl-2 proteins are subdivided into two groups: 1) the effector proteins Bax and Bak that contain 4 BH domains and 2) the BH3-only domain (Bad, Bid and BNIP3) contain only one BH domain [73,74]. In response to death stimuli, BH3-only Bcl-2 proteins activate Bax and Bak either directly or by inhibiting anti-apoptotic proteins. Bax and Bak then oligomerize at the OMM and induce MOMP and release of apoptotic factors, such as cytochrome c and the apoptosis-inducing factor.

Increases in cytosolic calcium in HF can induce apoptosis via several mitochondria-dependent and independent pathways. (1) Calcium-mediated activation of calpain cleaves Bid to its active form tBid and induces apoptosis in a Bax dependent manner [75]. (2) Elevated cytosolic calcium levels can also activate calcium dependent phosphatases, such as calcineurin. Calcineurin dephosphorylates Bad and activates the intrinsic mt-pathway [76]. (3) Massive calcium release from the ER/SR can directly trigger cytochrome c release from the mitochondria via direct calcium transfer into mitochondria via mitochondria associated membranes (MAM) that connects the ER to mitochondria. The most established MAM connection is the inositol triphosphate receptors (IP3R) at the ER level [77,78] and VDAC1 at the mt-level. GRP75 is a chaperone protein that stabilizes this tethering complex connecting the ER to mitochondria [79]. Thus, calcium released via IP3R in MAM can be rapidly up-taken into the mt-matrix via VDAC1 at the OMM and subsequently via MCU (most likely) or MCU-independent mode of calcium uptake at the IMM [80]. Depending on the intensity of calcium release, if intense enough to exceed the mt-matrix buffering capacity, calcium will bind to cyclophilin D and induces mt-permeability transition leading to mt-swelling, mt-vacuolar degeneration and release of cytochrome c [81,82]. 

Bcl-2 has been shown to regulate calcium homeostasis at the ER by directly interacting with the IP3R and decreasing IP3R mediated calcium release [83,84], Figure 2. Another mechanism by which the Bcl-2 family proteins regulate ER-mt calcium homeostasis and metabolism is via their interaction and modulation of VDAC, mainly VDAC1, which is the predominant VDAC isoform. Bcl-xL was shown to promote the open state of VDAC [85] and therefore prevent VDAC-mediated mt-calcium [86], while the BH3-only tBid and Bad were shown to promote the closed state of VDAC [87,88]. In its closed state, VDAC becomes permeable to calcium and sensitizes the mt-permeability transition pore to calcium [88]. The mt-death and mitophagy marker, BNIP3, has been shown to interact and to promote oligomerization of VDAC and mt-matrix calcium overload and dysfunction [48], Figure 4. The exact mechanism by which BNIP3 promotes VDAC oligomerization is not yet known. Moreover, BNIP3, through its transmembrane domain, has been shown to interact and inhibits the optic atrophy 1 (OPA1) activity, thereby promoting mt-fission and depressed mt-oxidative capacity [89,90]. BNIP3 also has been shown to selectively interact with nuclear and mt-encoded ETC complexes and to promote their degradation [91]. The MAPK, JNK, which is activated early on in cardiac stress, has been shown to repress the inhibitory effect of AKT on the transcription factor Forkhead box protein O3a (FOXO3a); thereby, inducing the transcription of FOXO3a effector BNIP3 gene [36] under cardiac stress condition. Moreover, JNK has been shown to phosphorylate Bcl-2 at Thr56/Ser70/Thr74/Ser87 [92] and inhibits it leading to Bax activation and dimerization at the OMM, OMM permeabilization, and apoptosis. Contrary to JNK, the mitochondria-anchored PKA phosphorylates Bad at Ser12 and inactivates it; thereby, promoting its cytoplasmic localization and cell survival in response to survival stimuli [93]. These data suggest that both the expression as well as the PTM of Bcl-2 family proteins to play an important role in the regulation of ER-mt calcium homeostasis, MOMP and apoptosis.

## 5. Signal Transduction Pathways Modulating Mitochondrial Dynamics and Function

Mitochondria play an active role in cellular signaling. While generation of ROS in mitochondria alters ROS sensitive signal transduction pathways, in turn altered cellular signaling can modulate mt-function to meet the ever-changing cellular environment and energetics-demand needs. In that regard, PTM of mt-proteins would be required to quickly adjust mt-function based on the rapidly changing energetic needs. This is because the induction of transcriptional and translational programs is a lengthy process and will not be feasible to serve the immediate need of the changing energetic demand. Proteomic work suggested that about 40% of the mt-proteome is phosphorylated [94,95,96,97]. This extent of mt-proteome phosphorylation is possible due to the presence of anchoring proteins, such as A-kinase anchoring protein 1 (AKAP1) and Sab, which enable the translocation and localization of PKA and JNK, respectively, to the OMM, Figure 2. Other kinases, such as PKCε and CaMKIIδ are also able to translocate to the mitochondria where they can also play an important role in modulating mt-function, Figure 2. AMPK is another important kinase that can translocate to the OMM and is known to regulate FA and glucose metabolism as well as myocardial biogenesis in the heart. In skeletal muscle (and likely in the heart), liver and adipose tissue, both PKA and AMPK, work synergistically to regulate key enzymes involved in glycogen metabolism, cholesterol synthesis and FA metabolism [98,99,100,101,102,103,104]. The calcium-sensitive serine/threonine phosphatase, calcineurin, is well known to be activated at the earliest stages of pathological hypertrophy [105], due to an increase in cytosolic calcium, and to antagonize PKA activity at the ER-mt interface [106,107,108], Figure 2 and Figure 4. Moreover, calcineurin plays important role in cardiac hypertrophy through the activation of calcineurin-NFAT signaling pathway [109]. Thus, calcineurin plays a detrimental role, while PKA exerts an overall cardioprotective effect in the heart, particularly when this relates to the regulation of ER-mt calcium homeostasis and function, calcium cycling and myocardial contractility and relaxation. The cyclic adenosine monophosphate (cAMP)-PKA signaling pathway is one of the predominant signaling pathways in the heart. Its regulation is diverse and complex, and differs from one cell compartment to another, depending on the regulation of AKAPs, phosphatases and phosphodiesterases (PDEs) activity in specific cellular sub-compartments, as discussed below, Figure 2. In the subsequent subsection, we will expand on important pathways and their role in modulating mt-function and metabolism in HF.

### 5.1. PKA

In the heart, cAMP is generated via the catecholamine-mediated stimulation of the β-adrenergic receptors. The cAMP modulates excitation-contraction coupling through the PKA-mediated phosphorylation of LTCC, RYR2, leading to CICR, necessary for initiation of the contractile phase of the cardiac cycle. Simultaneously, PKA phosphorylates cardiac muscle troponin I (TNNI3) allowing for binding of calcium to troponin C and faster force development and sarcomere shortening during systole [110], Figure 2. Moreover, PKA phosphorylates PLN at S16, thereby, enhancing SERCA2a activity and calcium pumping into the SR during diastole, Figure 2. In addition to the excitation-contraction coupling machinery, cAMP-PKA also phosphorylates a myriad of metabolic enzymes and transcription factors [111,112] in cardiac myocytes. The cAMP-PKA signaling is fine-tuned and tightly coordinated so that the appropriate signal is delivered without excessive PKA activation, which by itself would cause or worsen HF [113]. This is based on the theory or hypothesis that multiple and distinct cAMP pathways exist, which are spatially segregated in distinct subcellular compartments [114,115,116]. This hypothesis originated from the findings that isoproterenol infusion resulted in the activation of particulate fraction of PKA, whereas prostaglandin E1 increased the activity of soluble PKA [115]. It was found later on that the activation of PKA in specific subcellular compartments was related to PKA tethering to subcellular compartments by AKAPs [117]. PKA is a heterotetramer formed by two catalytic (C) subunits that are held in an inactive state by a dimer of regulatory (R) subunits, RI and RII. Binding of cAMP to the R subunits induces the dissociation of PKA C subunits and phosphorylation of PKA downstream targets. By anchoring PKA in very close proximity to its targets, AKAPs allow for a preferential phosphorylation of a local pool of PKA substrates [118]. Another important feature of AKAPs is their ability to gather and coordinate multiple other signaling proteins such as kinases, adenylyl cyclases (ACs), phosphatases and PDEs into a multifunctional transduction complex at different cellular subcompartments; thereby, allowing for multiple signals within discrete locals based on the local, subcellular, activity of the ACs, phosphatases and PDEs within the multifunctional complex [119]. Several AKAPs are expressed in cardiac tissue [120]. For example, AKAP18 isoform delta has been shown to anchor PKA to PLN and SERCA2a and to promote PKA phosphorylation of PLN in response to β-adrenergic stimuli in cardiac myocytes [121]. AKAP1 anchors PKA to the mitochondria. S-AKAP84, AKAP121, D-AKAP1 and AKAP149 are considered products of the single gene AKAP1, generated by alternative RNA splicing [122,123,124,125,126]. For further information on AKAPs and PKA regulation including different PKA, AC and PDE isoforms in the heart, the reader is directed to the review by Zaccolo [127].

In addition to its role in calcium cycling, PKA regulates enzymes related to glucose and FA metabolism, mt-dynamics, OXPHOS and Bcl-2 family of proteins as discussed earlier. PKA C, RI and RII subunits have been purified from the mitochondria [128]. These findings indicate that mt-targeted PKA can efficiently phosphorylate mt-proteins and protect the mitochondria from toxins and stress [129]. PKA has been shown to directly phosphorylate and positively affect the function of the ETC complexes I, IV and V, thereby improving mt-oxidative capacity and ATP synthesis [130,131,132,133,134,135]. PKA phosphorylation of the complex I subunit NADH:ubiquinone oxidoreductase subunit S4 (NDUFS4) at Ser173 facilitates its mt-import and its integration within complex I subunits by promoting its interaction with the Hsp70 chaperone protein [136]. NDUFS4 mutation at Ser173 residue is linked to complex I dysfunction in the mt-encephalomyopathy Leigh syndrome [137]. It has been shown that PKA also influences the stability of complex I by inhibiting proteases within the mitochondria that are capable of degrading complex I subunits NDUFA9, NDUFS4, and NDUFV2 [138]. In addition to its positive effect on mt-OXPHOS machinery, PKA was found to phosphorylate the glutathione S-transferase variant at Ser189 facilitating its transport into the mitochondria and enhancing mt-antioxidant capacity [139]. PKA modulates mt-dynamics through the phosphorylation of DRP1 at Ser637, thus promoting its cytoplasmic translocation [140]. Calcineurin exerts opposite effects and promote mt-fission [141]. As discussed earlier, PKA phosphorylates and inactivates Bad, thus making the mitochondria more resistant to apoptotic stimuli.

### 5.2. AMPK

AMPK is a heterotrimeric complex composed of an alpha and beta catalytic subunits (AMPKα and AMPKβ) and a gamma regulatory subunit (AMPKγ). AMPK is activated under conditions of energy stress and low ATP to AMP ratio. Binding of AMP to the gamma subunit causes a conformational change in AMPK, rendering it suitable for upstream kinases, such as liver kinase B1 (LKB1) or CaMK kinase, which phosphorylate AMPK at Thr172 and activates it [142,143,144,145,146]. AMPK has been shown to play an important role in mt-homeostasis by regulating mt-biogenesis and mt-dynamics, mt-quality control via mitophagy, and metabolism [147]. As noted earlier, AMPK is involved in regulating PGC-1α expression in concert with PKA. AMPK was found to phosphorylate the mt-fission factor at Ser155 and Ser172, which are required for the recruitment of DRP1 to mitochondria and induction of mt-fission in response to energy stress [148]. Moreover, AMPK phosphorylates ULK1 at Ser467, Ser555, Thr574 and Ser637 residues and activates autophagy during starvation and metabolic stress, thus promoting cell survival through mitophagy and removal of defective mitochondria [149,150]. The AMPKα_2_ subunit has been shown to be cardioprotective in cardiac stress by phosphorylating PTEN-induced putative kinase 1 (PINK1) at Ser495 promoting its recruitment to mitochondria and augmenting mitophagy and removal of defective mitochondria [151]. Phosphoproteomic work in skeletal muscle has identified mt-PKA scaffold protein, AKAP1, to be an AMPK target substrate. AMPK induced AKAP1 phosphorylation at Ser103 and enhanced mt-respiration in L6 myoblasts [152], likely via PKA dependent increase in mt-respiration and oxidative capacity as discussed earlier. Thus, AMPK is an essential kinase in the sensing and integration of cellular bioenergetic status and metabolism [153,154].

### 5.3. Other Kinases

The PKC isoforms are activated when biomechanical stressors or neurohormonal mediators activate the Gq/G11 receptor, which in turn activate phospholipase C (PLC) leading to the generation of inositol triphosphate (IP3) and IP3-mediated increases in cytosolic calcium and diacylglycerol (DAG). The four most functionally significant PKC isoforms in the heart are PKCα, PKCβ, PKCδ and PKCε. PKCα and β require calcium and DAG for their activation, while PKCδ and ε are activated by DAG only [155]. The activity of PKC isoforms is dependent on their expression level, their position/localization within the cellular compartment and their phosphorylation status [156]. PKCε is highly expressed in the heart [157,158] and is most abundant compared to the other PKC isoforms [159]. It is known for its ability to translocate to the mitochondria and to regulate mt-function and metabolism [160], in addition to its role in regulating cardiac contractile function via the modulation of sarcomeric proteins [161,162]. This is accomplished via the presence of receptors for activated C-kinase that anchor PKCε in close proximity to its target substrates [163]. There is a well-established concept that PKCε plays a cardioprotective role during ischemia via its preconditioning effect on the mitochondria [159,164,165,166]. Detailed proteomic work by Ping et al. demonstrated the presence of 36 proteins that form endogenous complex with PKCε [167], and that mt-proteins constituted an integral component of these complexes [167]. Moreover, PKCε was found to be present at the IMM and to be associated with mt-proteins involved in glycolysis, TCA cycle and FA-β-oxidation, OXPHOS and VDAC [168]. An unbiased proteomic approach has identified the mt-aldehyde dehydrogenase 2 to be one of the most important PKCε substrate involved in PKCε-mediated cardioprotection from ischemia [169]. Another study showed a direct interaction between PKCε and cytochrome c oxidase subunit IV (COXIV), and that preconditioning was associated with PKCε-mediated increase in COXIV phosphorylation and preservation of COXIV activity [170]. Subsequent phospho-proteomic work showed that PKCε phosphorylates ETC complexes I, II and III, and proteins involved in glycolysis, FA-β-oxidation, ketone body metabolism and heat shock proteins [171]. Through its interaction with the heat shock protein (HSP90), it was shown that PKCε was able to translocate and phosphorylate substrates at the IMM [172,173]. Moreover, PKCε may inhibit apoptosis through its interaction with and inhibition of Bad and Bax [174,175,176], and adenosine nucleotide translocase 1 (ANT1). Therefore, suggesting that PKCε may play a role in the regulation of mt-permeability transition pore, MOMP and apoptosis [160,171]. Collectively, these findings show that PKCε has a protective role under cardiac stress conditions by primarily preserving mt-function and metabolism and by inhibiting apoptosis.

In contrast to PKCε, JNK plays a maladaptive role in cardiac stress by primarily affecting mt-dynamics and apoptosis through the phosphorylation of MFN2 at Ser27 residue (promoting its degradation) [40], and the Bcl-2 family of proteins as discussed earlier. In the brain, JNK has been shown to phosphorylate and inhibit the E1α subunit of the pyruvate dehydrogenase (PDH) complex [177,178]; thereby, affecting pyruvate metabolism and prompting the increase in lactic acid. JNK may also phosphorylate and inhibit ETC complexes I [179,180] and III [181], either by phosphorylating them before or during mt-import [182] or by its translocation into the IMM. A previous work showed complex formation between ETC complex III and phosphorylated JNK and P38 [181]. Others have shown that overexpression of mt-aldehyde dehydrogenase 2 or AMPK is protective in the heart and endothelial cells, respectively, through the reciprocal attenuation of JNK activity and preservation of mt-function and biogenesis [183,184].

The serine/threonine kinase, CaMKII, is another kinase that is activated by increase in cytosolic calcium and may play a maladaptive role in the heart. CaMKII is activated upon binding of calcium to the calmodulin binding site of the regulatory domain leading to a conformational change, which exposes substrate-binding site of the kinase and the ATP binding site of the catalytic domain [185]. This leads to autophosphorylation of CaMKII at Thr287 residue and its activation. The CaMKII is also activated via its PTM by PKA [186] and is inactivated by calcineurin and PP1 through the dephosphorylation of Thr287 residue [187]. Furthermore, CaMKII is activated via oxidation of the Met281/282 residues [188], S-Nitrosylation of Cys290 via the nitric oxide-dependent pathway [189], and O-GlcNAcylation at Ser279 during hyperglycemic conditions [190]. The CaMKIIδ and γ isoforms are mainly expressed in the heart, with the CaMKIIδ outweighing the CaMKIIγ by about 2.5-fold [191]. Previous work showed that CaMKII activity was 3-fold increased [192] and CaMKIIδ expression was 2-fold increased in a failing human heart [193]. Moreover, there is a distinct CaMKIIδ PTM in the failing human hearts due to ischemic and non-ischemic etiologies. The CaMKIIδ phosphorylation at Thr287 residue was unaltered in ischemic, whereas it was increased in non-ischemic cardiomyopathy [194]. Moreover, oxidation of CaMKIIδ at Met281/282 was no different between ischemic and non-ischemic cardiomyopathy [194].

Previous work has shown that mt-targeted CaMKIIδ play a detrimental role through the phosphorylation of MCU at Ser57 and Ser92 with a net increase in MCU calcium entry and mt-matrix calcium overload; thus, promoting mt-dysfunction and apoptosis under cardiac stress conditions [55], Figure 4. Work in vascular smooth muscle cells showed that mt-CaMKIIδ induced MCU phosphorylation at Ser92 residue promoted mt-motility, vascular smooth muscle cells migration, and neointima formation [56]. Gq-mediated CaMKIIδ activation was found to be associated with mt-dysfunction and oxidative stress through the marked downregulation of the uncoupling protein 3 (UCP3) and PPARα. CaMKIIδ deletion or inhibition restored the expression of UCP3 and PPARα and significantly improved left ventricular function, cardiac fibrosis, apoptosis and ventricular arrhythmias [195]. Most recent work showed an increase in mt-CaMKII activity in failing mouse hearts one week after myocardial infarction surgery [196]. Myocardial and mt-CaMKII inhibition, via the expression of a potent and selective CaMKII inhibitor polypeptide (CaMKIIN), prevented left ventricular dilatation and dysfunction [196]. Transgenic expression of mt-CaMKII was associated with dilated cardiomyopathy, without myocardial concentric hypertrophy, due to reduced ETC complex I expression and mt-creatine kinase [196]. Genetic replacement of mt-creatine kinase by itself was able to rescue myocardial energetics, restore myocardial calcium homeostasis and prevent dilated cardiomyopathy [196]. All in all, these findings suggest a maladaptive role of mt-CaMKII signaling in cardiac stress via the dysregulation of mt-calcium homeostasis and mt-function.

## 6. Conclusions

In conclusion, mitochondria are central in the metabolic remodeling process and play an important role in the pathogenesis of HF. Signaling pathways and molecular mechanisms regulating ER-mt calcium signaling and function, as well as calcium cycling and metabolism are complex. Future work is needed to further understand the perturbed ER-mt relationship in HF and to elucidate molecular mechanisms and signaling pathways regulating ER-mt function and calcium homeostasis and their implications in the pathogenesis of HF. PKA signaling in the heart is by far the most complex in terms of its differential subcellular specific compartment regulation and of its ability to regulate calcium cycling, FA and glucose metabolism and mt-function. Overall cellular PKA signaling is well known to be downregulated in advanced HF. Whether differential PKA signaling (in)activation at subcellular specific compartments, particularly at the ER-mt interface exists in early systolic dysfunction, and prior to progression to overt systolic HF, need to be elucidated in future studies.

## Figures and Tables

**Figure 1 ijms-22-10579-f001:**
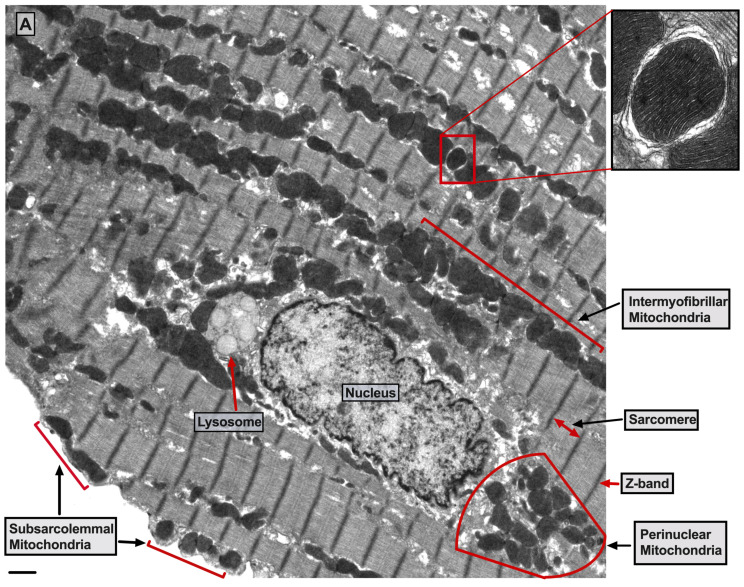
**Ultrastructure of normal myocardium.** (**A**) Transmission electron photomicrograph of adult cardiac myocyte showing perinuclear, intermyofibrillar and subsarcolemmal mitochondria. Red box in the figure is defining the zoomed image showing mitochondrion-containing autophagosome. Image 5k × magnified, scale bar 1 μm. (**B**) Transmission electron photomicrograph of adult cardiac myocyte showing the relationship of the intermyofibrillar mitochondria with the adjacent T-tubules and sarcomeres. Image 40k × magnified, scale bar 1 μm. Tracing in the zoomed image, defined by the black box, shows T-tubule (green line surrounding white area), junctional sarcoplasmic reticulum (yellow line with yellow shaded area), outer mitochondrial membrane (Blue line), inner mitochondrial membrane (pink line) and intermembrane space (purple shaded area).

**Figure 2 ijms-22-10579-f002:**
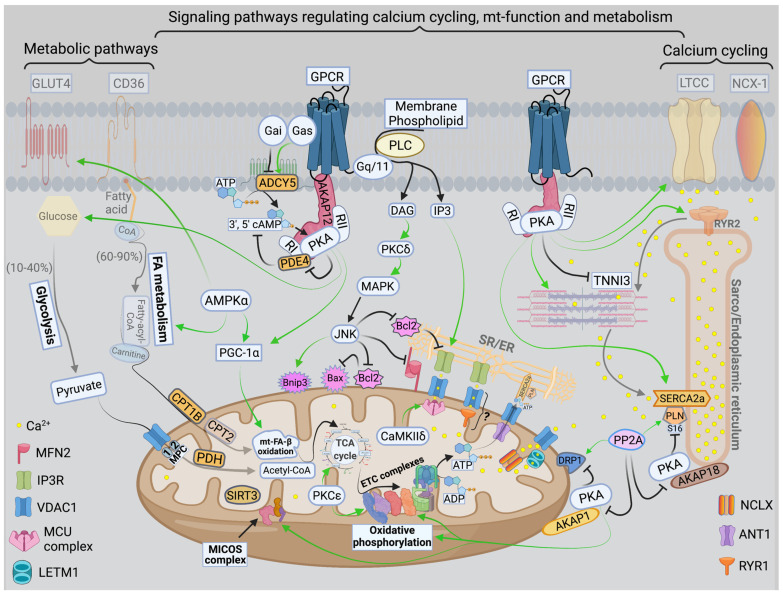
**Schematic drawing showing metabolic pathways, calcium cycling and implicated signaling in normal myocardium.** Green arrows promote signaling or activity. Metabolic pathways on the left show that in normal heart, fatty acid (FA) metabolism and mitochondrial (mt)-FA-β-oxidation contribute 60–90%, while glucose and pyruvate metabolism contribute 10–40% of the generated adenosine triphosphate ATP by oxidative phosphorylation (OXPHOS). On the far right is a presentation of calcium cycling and its regulation by protein kinase A (PKA) signaling. Please refer to text for details. In the center of the figure is a presentation of signaling pathways modulating mt-function and metabolism. The bottom center of the figure shows the tethering complexes that exist between the endoplasmic reticulum (ER) and neighboring mitochondria and how they crosstalk via calcium signaling. The presence of mt-calcium uniporter (MCU)-independent mt-calcium uptake via the ryanodine receptor 1 (RYR1) into the mt-matrix has not been clearly validated and is represented by a question mark (?). Abbreviations: GLUT4: glucose transporter member 4, CD36: FA transporter, MPC: mt-pyruvate carrier, PDH: pyruvate dehydrogenase, TCA: tricarboxylic acid, PKCε: protein kinase C epsilon isoform, PKCδ: protein kinase C delta isoform, CaMKIIδ: calcium/calmodulin kinase delta isoform, ADP: adenosine diphosphate, MFN2: mitofusin 2, IP3: inositol triphosphate, IP3R: inositol triphosphate receptor, VDAC1: voltage dependent anion channel 1, LETM1: leucine zipper and EF-hand containing transmembrane protein 1, NCLX: sodium/calcium/lithium exchanger, ANT1: ADP/ATP translocase 1, RYR2: ryanodine receptor isoform 2, SIRT3: sirtuin 3, MICOS: mt-cristae organizing system, ETC: electron transport chain, CPT1B: carnitine-O-palmitoyltransferase 1 muscle isoform, CPT2: carnitine-O-palmitoyltransferase isoform 2, AKAP: A-kinase anchoring protein, PP2A: calcineurin, DRP-1: dynamin-related protein 1, PLN: phospholamban, SERCA2a: sarco/endoplasmic reticulum calcium ATPase isoform 2a, TNNI3: troponin I, cardiac muscle, RI and RII: regulatory unit I and II, LTCC: L-type calcium channel, NCX-1: sodium/calcium exchanger 1, GPCR: G protein-coupled receptor, PLC: phospholipase C, DAG: diacylglycerol, MAPK: mitogen-activated protein kinase, JNK: c-Jun N-terminal kinase, Bcl-2: B-cell lymphoma 2, Bax: Bcl-2 associated X, BNIP3: Bcl-2 nineteen kilodalton interacting protein 3, AMPK: adenosine monophosphate-activated protein kinase, PGC-1α: peroxisome proliferator-activated receptor gamma coactivator-1 alpha. PDE4: phosphodiesterase 4, cAMP: cyclic adenosine monophosphate, and ADCY5: adenylate cyclase type 5.

**Figure 3 ijms-22-10579-f003:**
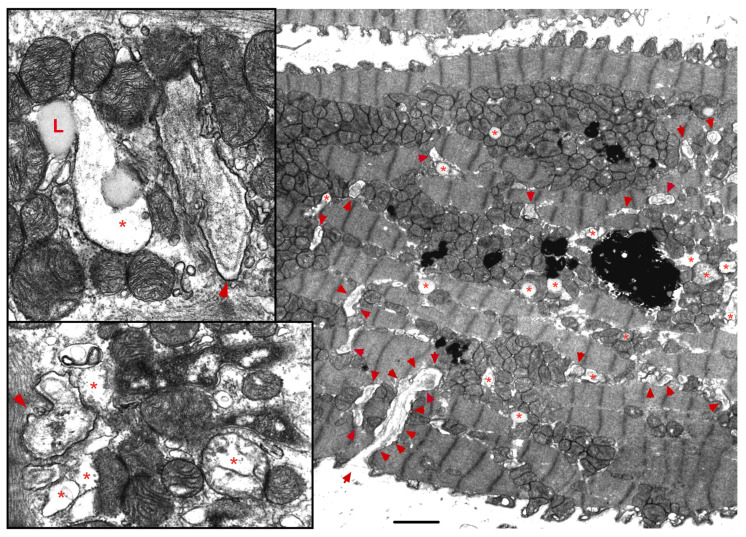
**Ultrastructure of the failing myocardium.** Transmission electron photomicrograph of myocardium from human failing heart showing typical clustering and fragmentation of intermyofibrillar mitochondria. Red asterisk showing mitochondria at advanced stage of vacuolar degeneration. There is perturbed relationship between the intermyofibrillar mitochondria and the surrounding sarcomeres and T-tubules (red arrowhead). Note that the T-tubules are dilated and that they are pushed to the edge of the clustered mitochondria, instead of being interspersed between adjacent mitochondria as seen in a normal myocardium. Moreover, numerous lysosomes (black, dense lamellar structures) are found interspersed between clustered mitochondria. Image 5k × magnified, scale bar 2 μm. Zoomed images showing in more detail the changes in mitochondrial cristae morphology and decrease in their density that occur in heart failure. Abbreviation: L: lipid.

**Figure 4 ijms-22-10579-f004:**
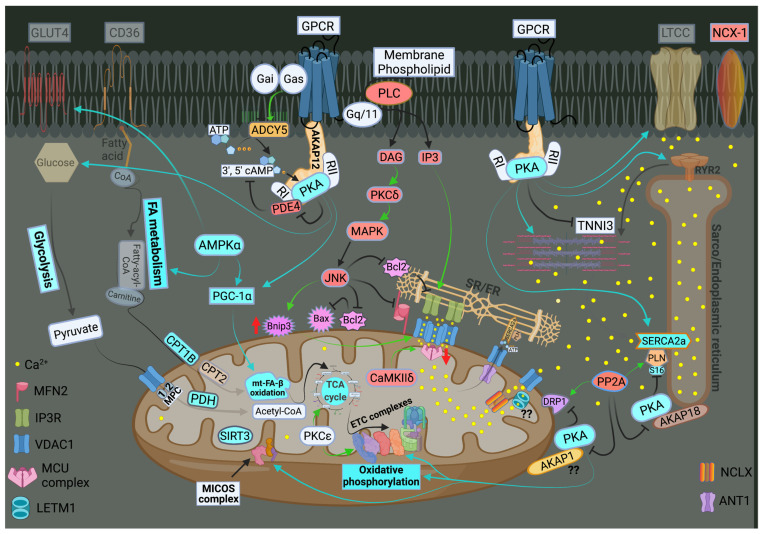
Schematic drawing showing metabolic remodeling, calcium cycling dysregulation and altered signaling in heart failure. Green arrows: promote signaling or activity. Turquoise arrows, rectangle or circle denotes decrease in signaling, activity or expression. Red rectangle or circle denotes increase in activity or expression. The deeper the intensity of the turquoise or red color, the higher is the degree in downregulation or upregulation, respectively, of protein activity or expression. Red up and down arrows indicate an increase in BNIP3 expression and a decrease in MCU expression, respectively. FA metabolism and mt-FA-β-oxidation, glycolysis, TCA and OXPHOS are attenuated in HF. SIRT3 expression is decreased in HF leading to mt-protein hyperacetylation. Enhanced MAPK (JNK) signaling inhibits Bcl-2 and MFN2 leading to ER-mt calcium dysregulation, mt-fission and apoptosis. Perturbations in calcium cycling and ER-mt tethering are evident in HF. ER calcium is depleted due to an enhanced ER calcium leak via RYR2, and a decrease in ER calcium uptake via SERCA2a leading to an increase in cytosolic calcium. Overall PKA signaling is attenuated in HF. Most importantly, decrease in PKA signaling and enhanced calcineurin activity at the ER-mt interface leads to (1) enhanced mt-fission, through recruitment of DRP1 to the mt-outer membrane; (2) decrease in ETC activity and OXPHOS; and (3) decreased in p-S16-PLN leading to further impairment in SERCA2a activity. Increases in IP3 abundance and BNIP3 expression lead to enhanced calcium release via the IP3R and enhanced calcium uptake via the VDAC1 channels, respectively. This is due to conformational change (oligomerization) of VDAC1. Enhanced CaMKIIδ expression or activity enhances mt-calcium uptake via MCU complex leading to mt-matrix calcium overload and mt-dysfunction. Mitochondrial calcium efflux is likely to be affected as well. It is unclear whether there is post-translational modification (??) and decrease in activity or expression of LETM1 and AKAP1 in HF. Abbreviations are listed under Figure 2.

## Data Availability

Not applicable.

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
