# Peer review of "Metabolic Remodeling and Implicated Calcium and Signal Transduction Pathways in the Pathogenesis of Heart Failure"

_ijms, 2021, doi:10.3390/ijms221910579_

Round 1

Reviewer 1 Report

Please see the attached file for my comments.

Author Response

Reviewer 1:

Overall, although this is a comprehensive review, it is not “deftly” written. In general, the author needs to attend to correcting punctuation throughout.

We have proofread the manuscript and did the corrections that the reviewer suggested. We also made additional corrections to typo mistakes or punctuations that we have missed.

If the reviewer or the editors think that the manuscript requires professional proofreading and editing, we are happy to do so.

There are many acronyms used, which at times distracts the reader. I would suggest that some that are not used often could be typed out in full, with a list of others provided for ready access to the readers.

We have done so. Abbreviations that were used once or twice were removed and a list of frequently used abbreviations was included as a separate table on the last page of the revised manuscript.

Endoplasmic reticulum (ER) and Sarco/endoplasmic reticulum (SR) are used frequently throughout, yet I am unclear as to whether these are separate organelles, or one and the same? Can you clarify please?

The SR and ER are the same organelles except for the junctional SR which contains the ryanodine receptors and is found at a closed proximity with the plasma membrane.

Line 13. Sentence beginning “This is due to their topology….” Needs rewriting, or at least punctuation, to improve meaning.

This change has been done.

Line 17. Remove comma after “both” in sentence ending “with signaling pathways

regulating both, calcium cycling and mitochondrial function”.

This change has been done.

Line 19. Final sentence needs punctuation.

This change has been done.

Line 26, “powerhouse” should be plural.

We thank the reviewer for his suggestion, but we respectfully disagree.

Line 27, “extend” should be plural.

This change has been done.

Lines 32- 35, claims “it is well known that” ……are impaired at the earliest stages of cardiac remodeling……with progression to overt SHF”. This statement references just one other Review. I would expect the original literature to be cited after such an important statement.

We have done so. We thank the reviewer for his great suggestion.

There are many other unreferenced statements in this section, which is not what I would expect of a quality Review article.

We thank the reviewer for his suggestion. We have added references to statements in the “Introduction” part of the manuscript. For PKA, PKC, Calcineurin etc… work we did not cite references as they were extensively discussed later on in the body of the manuscript. However, we added the following statement: “as discussed later in detail under section 5 of the manuscript” on page of the manuscript. Please refer to the tracked version of the manuscript.

Line 32 – 34, sentence beginning “To date, it is…..” badly needs punctuation!

We have made extensive revision and correction to this section of the manuscript.

Line 35-36, sentence beginning ‘Glucose metabolism….” Makes no sense as written.

We have made appropriate correction as highlighted in the Tracked version of the manuscript.

Lines 71 – 83. I assume this is referring to cardiomyocytes in this section, and not the “heart” as stated in line 71? Please be aware that cardiomyocytes are not the most abundant cells in the heart! Importantly, other cells also contribute to signaling pathways.

We do agree with the reviewer and thank him/her for the comment. We have replaced heart with cardiac myocyte.

Line 76. Needs rewording. What does “runs in” mean here? “The longitudinal SR runs in like a network around the intermyofibrillar mitochondria…”

We have replaced “runs in like” with “forms”.

I dislike the use of “so-called” when referring to CICR, it sounds dismissive.

We do apologize if this statement was perceived as dismissive, which is not our intention. We have replaced “so-called” with “CICR phenomenon”.

Lines 120 – 123 describes the process of CICR, yet does not reference the work of Fabiato and Fabiato! Instead, there are inappropriate refs (in my mind) to modeling papers. Please remedy this.

We have cited the work by Fabiato et al.

Line 126. Reference 14 is rather obscure for regulation of SERCA by PLN! Please cite the original work detailing the role of PLN in SERCA function.

We have done so.

Line 127. I don’t agree that “The SR/ER and the mitochondria are considered the hub of calcium cycling.” Please replace with a more appropriate statement, e.g.“are an important component of myocyte calcium cycling”.

We have performed the change as suggested.

Line 133. What do you mean by “the mostly abundant voltage dependant anion channel”? Please clarify.

Out of the three VDAC isoforms, VDAC1 is the most abundant. We have clarified this in the manuscript as well. Please refer to tracked changes version of manuscript.

Line 143. Change “eluted” to “eluded” in sentence “Moreover, recent investigation has eluted to 143 the importance of the hydrogen/calcium exchanger…”

Done.

I found this section (Bcl-2) difficult to follow. Such a lot of acronyms! Can the section be simplified? There does seem to be too much detail in my opinion.

We simplified this section. Please refer to the tracked changes version for sections that were deleted.

Lines 306 – 308. Can you clarify what is meant by “Massive calcium release….”? Is this not what happens during each calcium transient? Hard to think that there would be an even greater calcium release in cardiac myocytes unless they were massively calcium overloaded and irreversibly damaged.

On beat-to-beat basis under normal conditions, SR calcium release is about 60-70% of SR calcium content. Massive calcium release can happen under high catecholaminergic states, such as Takotsubo cardiomyopathy, likely through enhanced release via the IP3 receptors.

Lines 386 – 389. This needs to be written as 2 sentences.

This has been done.

Line 420. Use an uppercase Z” for Zaccolo.

Done.

The remaining recommended changes were addressed as well.

Figures 1 & 3 are both very dark images. I’d expect better quality EMs that clearly show the cristae within mitochondria. Fig 1 B claims to show t-tubules (yellow lines), but I find this unconvincing. What is the large white filled structure? Fig 3 is similarly of poor quality. For review purposes, I’d like to see evidence that the ultrastructural changes really were typical of failing myocardium. EM of isolated cardiac myocytes produces processing artifacts.

Alternatively, please reference examples from the literature showing the extensive

ultrastructural changes described for failing myocardium.

We have increased the brightness of figures 1 & 3. Figure 1B was replaced by another figure where the cristae of the mitochondria are better seen and the T-tubule is more demarcated.

We also modified figure 3 to include zoomed images where the mitochondrial cristae and dilated t-tubules are better visualized. I hope that these changes are satisfactory to the reviewer.

Figure 2. There is a lot to take in here. Could it be simplified? For example, is it necessary to include EC coupling (RHS), given that it has little interaction with the mitochondria in the figure? The hollow yellow circles representing Ca2+ need to be a darker colour. There are also two differently drawn organelles representing the SR/ER which is confusing.

We have simplified figure 2 by making the EC coupling machinery, FA and glucose metabolism pathways less apparent by decreasing their opacity. Also, we made the background slightly darker and made the small yellow dots brighter so that they are more apparent. I hope that these changes would satisfy the reviewer.

We purposely used two organelles representing SR/ER. The one close to the mitochondria representing SR/ER as a network and associated ER-mt tethering complexes. The tubular looking SR/ER is to show the longitudinal SR and the junctional SR that is in close proximity with the LTCC.

Figure 4. Again, this is a difficult figure to interpret. The turquoise (decrease) and red (increase), plus intensity (degree) combinations is a good idea in theory, but it was difficult to detect intensity differences on my screen.

We made the background darker so that these changes are more apparent. Moreover, although the images are of high quality when they are inserted into the word document, it looks like they loose some quality. Original images will be uploaded upon resubmission of the manuscript.

Reviewer 2 Report

In this narrative review, Dr. Chaanine discussed the roles of mitochondrial calcium handling (i.e., cycling and signaling) in heart failure. Overall, this is a very detailed and comprehensive review, which would highlight the significance of mitochondria in cardiomyocytes and could potentially be useful for future research in the field. Nonetheless, I do have some suggestions to be incorporated in the manuscript:

  • The title needs to be more specific. The words "mitochondria" and "calcium" need to be specified here.
  • Instead of discussing the metabolic / molecular / calcium changes in HF as a general term, I think the author needs to differentiate the changes (e.g., metabolic remodeling, calcium cycling dysregulation and altered signaling) observed in HFrEF and HFpEF. By explaining the specific changes on the molecular remodeling based on the clinical phenotypes, the manuscript would be more insighful and up to date. 
  • I am wondering if there is any known interaction between nuclear and mitochondrial calcium handling? If so, perhaps the discussion on that particular topic could be included. 
  • Line 26: Perhaps the total number of mitochondria in cardiomyocytes could be added and compared with other organs to emphasize that indeed cardiomyocytes have the highest number of mitochondria in human body.
  • Line 56-58: The word "heart failure" is missing from this sentence. It is needed since heart failure is the focus of this review, as exemplified from the title.
  • I think in general, this publication about the roles of cardiomyocyte calcium handling in human heart needs to be added in the manuscript (PMID: 32188566).
  • Perhaps there is a good way to combine Figure 2 and 4? At the moment, it is difficult to see which components of the signaling cascades are altered in heart failure. Perhaps combining the two figures would help to clearly depict the disease-related changes even more.
  • Please confirm whether the author has obtained permission to reproduce or display all the figures (if they were taken from somewhere else). 

Author Response

Reviewer 2:

The title needs to be more specific. The words "mitochondria" and "calcium" need to be specified here.

We have changed the title to “Metabolic remodeling and implicated calcium and signal transduction pathways in the Pathogenesis of Heart Failure”. The term Metabolic remodeling is a broad term that includes mitochondrial dysfunction and derangements in metabolism.

Instead of discussing the metabolic / molecular / calcium changes in HF as a general term, I think the author needs to differentiate the changes (e.g., metabolic remodeling, calcium cycling dysregulation and altered signaling) observed in HFrEF and HFpEF. By explaining the specific changes on the molecular remodeling based on the clinical phenotypes, the manuscript would be more insightful and up to date.

We thank the reviewer for raising this important comment. Although it would be interesting to compare HFpEF with HFrEF, it is technically difficult to do so because: 1) It will make the paper more complex and challenging to write, 2) HFpEF is more of a heterogeneous disease and 3) Unlike HFrEF, no good animal models of HFpEF exist that recapitulate the human disease.

We tried to highlight changes in HFpEF vs HFrEF when these changes are reported in literature such as the case of oxidative phosphorylation. But it is extremely challenging to compare changes in signaling between HFpEF and HFrEF.

I am wondering if there is any known interaction between nuclear and mitochondrial calcium handling? If so, perhaps the discussion on that particular topic could be included.

We are not aware that there are any known interactions between nuclear and mt-calcium handling. We performed a literature search and did not find any published work on this topic.

Line 26: Perhaps the total number of mitochondria in cardiomyocytes could be added and compared with other organs to emphasize that indeed cardiomyocytes have the highest number of mitochondria in human body.

We have cited the work by D'Erchia et al. (citation #2) that compares mt-content between different organs.

Line 56-58: The word "heart failure" is missing from this sentence. It is needed since heart failure is the focus of this review, as exemplified from the title.

We have done so.

I think in general, this publication about the roles of cardiomyocyte calcium handling in human heart needs to be added in the manuscript (PMID: 32188566).

We have done so.

Perhaps there is a good way to combine Figure 2 and 4? At the moment, it is difficult to see which components of the signaling cascades are altered in heart failure. Perhaps combining the two figures would help to clearly depict the disease-related changes even more.

We have modified both figures to make changes in signaling more apparent: 1) made the background darker, 2) EC calcium handling proteins and FA and glucose metabolism pathways were made less visible so that changes in signaling and mt-changes are more apparent, and 3) The yellow dots representing calcium were made brighter so that they are more apparent.

Please confirm whether the author has obtained permission to reproduce or display all the figures (if they were taken from somewhere else).

All figures are original and are not taken from anywhere else.

Round 2

Reviewer 1 Report

I am happy that the points raised during my first review of this manuscript have been addressed. I am not sure that Figures 2 & 4 have been improved by changing the background colour however. I will leave that to the Editors to decide.

Reviewer 2 Report

Thank you for the responses to my previous comments. I have no further remarks.